# Molecular Characteristics, Circularization Mechanism, and Potential Functions of circRNA-0172

**DOI:** 10.3390/genes16111282

**Published:** 2025-10-29

**Authors:** Haojie Wang, Wenhao Li, Ting Yang, Hao Bai, Lingling Qiu, Guobin Chang

**Affiliations:** 1College of Animal Science and Technology, Yangzhou University, Yangzhou 225009, China; hjwang2024@163.com (H.W.);; 2Joint International Research Laboratory of Agriculture and Agri-Product Safety, The Ministry of Education of China, Institutes of Agricultural Science and Technology Development, Yangzhou University, Yangzhou 225009, China

**Keywords:** chicken, circRNA-0172, circularization mechanism, ceRNA

## Abstract

Background/Objectives: Avian leukosis virus subgroup J (ALV-J) harms poultry via tumors and immunosuppression; MDM2-derived circRNA-0172′s role in ALV-J infection is unknown. This study explores its molecular characteristics, circularization mechanism and functions for ALV-J control. Methods: CEF/DF-1/HD11 cells were cultured; RNA/DNA was extracted. circRNA-0172 was validated via PCR, sequencing and RNase R; subcellular localization and circularization were analyzed with fractionation and vectors. miRanda/TargetScan/GO/KEGG, CCK-8 and qRT-PCR assessed targets, pathways and functions. Results: circRNA-0172 (1180 bp, MDM2 exons 3–11) is cytoplasm localized, RNase R resistant, and circularized via flanking ERVL-MaLR repeats. Highly expressed in immune tissues/HD11 cells, it is downregulated by ALV-J. Its overexpression inhibits DF-1 proliferation, downregulates Bcl2 and upregulates CCND1. Conclusions: This is the first confirmation of circRNA-0172′s stability and repeat-dependent circularization; it regulates cell proliferation/apoptosis, providing potential targets for ALV-J-associated avian leukemia.

## 1. Introduction

Circular RNAs (circRNAs) belong to the class of non-coding RNAs, though a small number can encode functional peptides, generated by reverse splicing of precursor mRNA (pre-mRNA), forming closed circular transcripts through reverse covalent linkage of free ends [1,2,3]. They typically exhibit low-level expression and cell/tissue specificity [4]. CircRNAs are widely present in eukaryotic cells (e.g., animals, plants, fungi) [5]. Unlike linear RNAs (e.g., mRNA, tRNA), they lack free 5′ caps and 3′ poly (A) tails, making them more stable and less prone to nuclease degradation [6].

The functions of circRNAs are diverse and complex, primarily involving competing endogenous RNA (ceRNA), gene expression regulation, and translation involved in cellular physiological and pathological processes [7]. For instance, Sebastian et al. showed that expressing human CDR1as in zebrafish disrupts midbrain development, mimicking the phenotypic characteristics of miR-7 downregulation, thereby first identifying *CDR1as* as a potential ceRNA for miR-7 [8]. Chen et al. found that circHIPK3 adsorbed miR-124 in hepatocellular carcinoma through ceRNA action, deregulated the inhibition of *AQP3* by miR-124, thus upregulating *AQP3* expression, and ultimately promoted tumor cell proliferation and migration [9]. Abdelmohsen et al. discovered that circPABPN1 binds to the HuR protein, preventing HuR from interacting with *PABPN1* mRNA and reducing *PABPN1* translation, first uncovering a mechanism where circRNAs compete with homologous mRNAs for RNA-binding proteins [10]. Lesca et al. first reported that circANRIL binds to the nucleolar protein PES1, inhibiting exonuclease processing of precursor rRNA (e.g., 36S and 32S), blocking ribosome biogenesis, inducing nucleolar stress response and activation of p53, and inducing apoptosis while suppressing macrophages and vascular smooth muscle cells to slow atherosclerosis progression [11]. Li et al. showed that EIciRNA (exon-intron circRNA) cis-enhances host gene transcription by binding to RNA Pol II and U1 snRNP [12]. Ivano et al. first confirmed functional translation of endogenous circRNAs in eukaryotic cells, challenging the traditional “non-coding” perception [13]. Yang et al. discovered that circ-FBXW7 is translated into the FBXW7-185aa short peptide, which inhibits tumor growth by promoting the degradation of the c-Myc oncoprotein [14]. CircRNAs regulate gene expression at multiple levels (transcription, translation, epigenetics) and deeply participate in disease development. Their multifunctionality makes them a frontier in life science research and offers new directions for clinical diagnosis and therapy (e.g., circRNA-targeted drugs, exosome delivery systems). With technological advancements, more hidden functions of circRNAs will be revealed, driving precision medicine forward.

In our previous work, we conducted whole-transcriptome sequencing of spleen tissues from ALV-J-infected black-bone silky fowls and identified circRNA-0172 as a differentially expressed circRNA, leading us to hypothesize its critical involvement in ALV-J infection. ALV-J is a retrovirus that severely threatens global poultry health, mainly infecting chickens to cause myelocytomatosis and other tumors, as well as immunosuppression that raises secondary infection risks and reduces production performance. circRNA-0172 is formed by circularizing 9 exons (exons 3–11) of the murine double minute 2 (*MDM2*) gene. The *MDM2* gene is a key proto-oncogene, and its encoded *MDM2* protein is vital for cell growth, proliferation, and tumorigenesis [14,15,16,17].

Under normal physiological conditions, *MDM2* acts as an E3 ubiquitin ligase, recognizing and binding to p53 to ubiquitin tag it for proteasomal degradation. This maintains low p53 levels in normal cells, ensuring proper proliferation and division [18,19,20]. Upon cellular stress (e.g., DNA damage, oxidative stress, nutrient deficiency), p53 is activated (e.g., via phosphorylation), inhibiting *MDM2*-p53 binding and allowing p53 accumulation. Activated p53 then initiates downstream pathways: inducing cell cycle halt to allow DNA repair, facilitating DNA repair mechanisms, or triggering apoptosis if repair fails, thus preventing cancerous transformation [21,22,23,24]. Therefore, *MDM2* and p53 form a negative feedback loop to maintain genomic stability.

In pathological states (e.g., cancer), *MDM2* is often overexpressed due to chromosomal amplification or mutations in various tumors (e.g., sarcomas, breast cancer, liver cancer, neuroblastoma) [25,26,27,28]. Excess *MDM2* continuously degrades p53, disabling its tumor-suppressive functions and allowing cells to evade growth control, proliferate uncontrollably, and resist apoptosis, promoting tumor formation [29,30,31]. *MDM2* abnormalities typically rely on tumor cells with wild-type (unmutated) p53. If p53 itself is mutated (as in more than 50% of human tumors), tumors may escape regulation through other pathways, but *MDM2* abnormalities can still synergize with other mechanisms to accelerate cancer progression [32,33,34,35,36]. *MDM2* overexpression typically correlates with high tumor malignancy, poor prognosis, and resistance to chemotherapy/radiation (due to suppressed p53-mediated apoptosis) [37,38,39].

This study aims to characterize circRNA-0172 molecular features, predict its regulatory mechanisms, and lay a foundation for exploring its roles in ALV-J infection.

## 2. Materials and Methods

### 2.1. Cell Isolation and Culture

chicken embryo fibroblast (CEF) cells Culture: Embryos were removed from 10 day eggs and transferred to a sterile Petri dish with 5 mL PBS (solarbio, code no. P1020, Beijing, China). Muscle tissues rich in fibroblastic tissue from the trunk and wings were dissected, placed in a new dish, and digested with 5 mL pre-warmed (37 °C) 0.25% trypsin-EDTA (Gibco, code no. 25200072, Grand Island, NY, USA) for 15–20 min (gently shaken every 5 min). Digestion was terminated with an equal volume of complete medium containing serum. Cell suspensions were gently pipetted to dissociate cells, transferred to 15 mL centrifuge tubes, and centrifuged at 1000 rpm for 5 min at room temperature. The pellet was resuspended in 5 mL complete medium, filtered through a 200-mesh cell strainer, and viable cells were counted using Trypan Blue staining and a hemocytometer. Approximately 1 × 10^6^ cells were plated into 25 cm^2^ culture flasks and incubated at 37 °C with 5% CO_2_.

DF-1 and HD11 Cell Culture: DF-1 (acquired from the American Type Culture Collection, code no. CRL-3586, Manassas, VA, USA) and HD11 cell cryogenic storage tubes were rapidly thawed in a 37 °C water bath, transferred to centrifuge tubes with 5 mL complete medium, centrifuged at 1000 rpm for 5 min, and resuspended in fresh medium. Cells were cultured at 37 °C with 5% CO_2_ and 60–70% humidity in DMEM (Cytiv, code no. SH30243.01, Marlborough, MA, USA) supplemented with 10% FBS (Gibco, code no. 10099141C), 1% penicillin-streptomycin (Gibco, code no. 15140122, Waltham, MA, USA), and GlutaMAX (Thermo Fisher Scientific, code no. PB180419, Waltham, MA, USA).

### 2.2. Total RNA Extraction and Quality Detection

Tissues (heart, liver, spleen, lung, kidney, lymph, bursa fabricius) from three adult roosters (provided by Jiangsu Lihua Animal Husbandry Co., Ltd., Changzhou, China) were stored at –80 °C for total RNA extraction using TRIzol^®^ (Thermo Fisher Scientific, code no. 15596026CN). RNA purity and concentration were measured via Nanodrop ND-1000 (Thermo Fisher Scientific), with ideal A_260_/A_280_ ratios of 1.8–2.0 and A_260_/A_230_ ≥ 2.0.

### 2.3. Genomic DNA Extraction

Genomic DNA from CEFs was extracted using the Tiangen Blood/Cell/Tissue Genomic DNA Extraction Kit (TIANGEN, code no. DP-304, Beijing, China), and purity/concentration were assessed via Nanodrop ND-1000 (ideal A_260_/A_280_ ratios of 1.8–2.0).

### 2.4. Validation of circRNA-0172

Convergent and divergent primers (Table 1) were constructed according to MDM2 reference sequences (NCBI accession: NM_001199384.2) and circRNA-0172 sequences to verify its existence and splicing pattern. PCR was performed using CEF genomic DNA (gDNA) and cDNA, with products sequenced by Qingke Biotech (Beijing, China).

### 2.5. Quantitative Real-Time PCR (qRT-PCR) Analysis

RNA was reverse-transcribed to cDNA using HiScript IV RT Super Mix for qPCR (TaKaRa, code no. R433-01, Kyoto, Japan) with gDNA wiper. qRT-PCR was performed using Taq Pro Universal SYBR qPCR Master Mix (Vazyme, cat. no. Q712-02, Nanjing, China) on a QuantStudio™ 5 real-time PCR system (Thermo Fisher Scientific), with GAPDH as the internal reference gene. Relative expression was calculated via the 2^−ΔΔCt^ method. Primers were designed using Primer 5.0 (PREMIER Biosoft, San Francisco, CA, USA) and Oligo 7 (Molecular Biology Insights, San Francisco, CA, USA) synthesized by Qingke Biotech (Table 1), with triplicate experiments.

### 2.6. Subcellular Localization

Nuclear and cytoplasmic fractions were isolated with NE-PER Nuclear and Cytoplasmic Extraction Reagents (Thermo Fisher Scientific, catalog no. 78835). TRIzol was used to extract total RNA from whole-cell lysates, tissues, or fractions, and qPCR was applied to analyze circRNA-0172 expression in the nucleus and cytoplasm.

### 2.7. RNase Digestion Assay

Total RNA (5 μg) was divided into two groups: experimental (3 U/μg RNase R) (GENESEED, code no. R0301, Guangzhou, China) and control (equal volume RNase-free water) (solarbio, code no. IR9123, Beijing, China). Samples were incubated at 37 °C for 30 min, heat-inactivated at 70 °C for 10 min, reverse-transcribed to cDNA using HiScript IV RT Super Mix (TaKaRa, code no. R433-01, Kyoto, Japan), and qRT-PCR was used to detect relative expression of circRNA-0172 and its host gene *MDM2* and validate circular structure and stability of circRNA-0172.

### 2.8. Vector Construction

Recombinant cloning was used to generate the overexpression vector pcDNA3.1-circRNA-0172. Briefly, primers (sequences listed in Table 1) were designed using SnapGene v7.2 (SnapGene, San Diego, CA, USA) and synthesized by Beijing Qingke Biotechnology. PCR amplification was performed with cDNA or gDNA as templates. Following agarose gel electrophoresis, PCR products were purified using the TaKaRa MiniBEST Agarose Gel DNA Extraction Kit ver. 4.0 (TaKaRa, code no. 9762). The pcDNA3.1 plasmid was selected as the vector backbone, linearized by digestion, and then subjected to homologous recombination with the purified target fragment using the ClonExpress Ultra One Step Cloning Kit (Vazyme, code no. C115-01). Recombinant products were transformed into *Escherichia coli* (*E. coli*) competent cells (Qingke Biotech, Code No. TSC-C01), and positive clones were initially screened by colony PCR after shaking culture. Validated positive clones were confirmed by Qingke Biotech Sanger sequencing, and plasmids were extracted by the EndFree Mini Plasmid Kit II (TIANGEN, Code No. DP118) and characterized for concentration and purity. Finally, validated overexpression vectors were generated for subsequent functional assays.

### 2.9. Cell Transfection

Depending on experimental requirements (e.g., 6-well or 12-well plates), DF-1 cells were plated one day prior to transfection to ensure that cell confluency ranged between 50% and 70% at the time of transfection. Using lipofectamine™ 2000 (Thermo Fisher Scientific, code no. 11668019) transfection reagent, plasmid DNA was mixed with the transfection reagent according to the manufacturer’s instructions and added to the cells. After 4–6 h of plasmid DNA transfection, replace with fresh medium to reduce the cytotoxicity caused by transfection reagents. ALV-J (Courtesy of Yangzhou University College of Veterinary Medicine) infected target cells at MOI = 2. Virus was thawed on ice, diluted per MOI calculation, added to wells after PBS washing, and incubated at 37 °C/5% CO_2_ for 2 h. Unbound virus was removed; 1% serum medium without dual antibodies was added. Each group had 3 biological replicates. At 3–96 h post-infection, cells were imaged via inverted microscope, followed by downstream analyses.

### 2.10. Cell Viability Assay

Cell proliferation and viability were assessed via CCK-8 kit (Vazyme, code no. A311-01). Cells were plated into 96-well plates, pre-incubated at 37 °C, and divided into blank (medium only), control (untreated), and experimental groups. After treatment (24–72 h), 10 μL CCK-8 reagent was added to each well, incubated for 1 h, and absorbance was measured at 450 nm. Cell viability (%) = [(Experimental—Blank)/(Control—Blank)] × 100.

### 2.11. Bioinformatics Analysis

Genome analysis was performed using Ensembl (https://www.ensembl.org/index.html?redirect=no, accessed on 5 January 2025). MiRNA targets and their corresponding target genes of circRNA-0172 were predicted using miRanda (http://mirtoolsgallery.tech/mirtoolsgallery/node/1055, accessed on 5 May 2025), DIANA TOOLS (http://diana.imis.athena-innovation.gr/DianaTools, accessed on 6 May 2025), and TargetScan Human 8.0 (https://www.targetscan.org/vert_80/, accessed on 6 May 2025). ceRNA networks were constructed and visualized via Cytoscape 3.10.3. GO functional enrichment and KEGG pathway analysis were performed via Kidio Cloud Platform (https://www.omicshare.com/, accessed on 10 May 2025)

### 2.12. Statistical Analysis

Data are presented as the mean ± standard error of the mean (SEM). Statistical analyses were performed via SPSS 26.0 (SPSS Inc., Chicago, IL, USA) and GraphPad Prism 10 (GraphPad Software Inc., San Diego, CA, USA): independent samples *t*-test for two-group comparisons, one-way ANOVA for multi-group comparisons, and Mann-Whitney U test for non-normally distributed data. All experiments included triplicates, with significance defined as (* *p* < 0.05, ** *p* < 0.01, *** *p* < 0.001, **** *p* < 0.0001).

## 3. Results

### 3.1. Characterization of Chicken circRNA-0172

Based on our previous transcriptome dataset (GSE118752) from ALV-J-infected black-bone silky fowls spleens [40], circRNA-0172 was differentially expressed between ALV-J-infected and non-infected groups. Host gene MDM2 mRNA encodes MDM2 protein, which regulates the p53 signaling pathway in cell cycle, apoptosis, and tumorigenesis and was therefore selected for validation. Ensembl genomic analysis revealed circRNA-0172 is an exonic circRNA formed by backsplicing *MDM2* exons 3–11, spanning 1180 bp (Figure 1A). Divergent primer amplification yielded the expected product, verified by Sanger sequencing (Figure 1B). Divergent primers amplified circRNA-0172 only from cDNA, while convergent primers amplified linear *MDM2* from both cDNA and gDNA (Figure 1C), providing preliminary evidence that this RNA is a circular molecule. The stability of circRNA-0172 was further verified by RNase R exonuclease digestion assay. Results showed that compared with the control group, circRNA-0172 showed no significant difference, whereas the corresponding linear transcript *MDM2* mRNA was significantly degraded (Figure 1D). Further description of circRNA-0172 as a circRNA with a stable cyclic structure.

Subsequently, the expression patterns of circRNA-0172 in different cell lines and chicken tissues were analyzed. Among the three tested cell lines, both circRNA-0172 and its host gene MDM2 mRNA showed the highest relative expression in HD11 cells (Figure 1E). Tissue expression profiling indicated that circRNA-0172 was highly relative expression expressed in the spleen, lung, lymph gland, and bursa fabricius, while *MDM2* mRNA showed highly relative expression in the lung, lymph gland, fallopian tube, and dorsal skin (Figure 1F), suggesting they may critical roles in functional regulation of immune cells and immune response mechanisms. Subcellular localization analysis of circRNA-0172 and *MDM2* mRNA revealed that circRNA-0172 had significantly higher relative abundance in the cytoplasm than in the nucleus, implying a potential functional role through the ceRNA mechanism, whereas *MDM2* mRNA was predominantly distributed in the nucleus (Figure 1G).

To further understand the effects of ALV-J on circRNA-0172, we performed qRT-PCR on ALV-J transfected DF-1 cells at different timepoints to measure the relative expression levels of the viral gene (*Env*), circRNA-0172 and *MDM2*. The results revealed a significant upregulation of *Env* expression post-infection (Figure 1H), confirming successful ALV-J infection in DF-1 cells. Concurrently, compared with the control group, the expression levels of circRNA-0172 (Figure 1I) and *MDM2* (Figure 1J) were downregulated over the course of infection, indicating a time-dependent association between ALV-J infection and the suppression of these host genes.

Taken together, this study demonstrates that circRNA-0172 is a circular RNA stable in structure that shows differential expression in different chicken tissues and related to ALV-J infection.

### 3.2. Circularization of circRNA-0172 Depends on Flanking Sequences

To investigate the circularization mechanism of circRNA-0172, we first predicted the repetitive features of its flanking intronic sequences using RepeatMasker (http://www.repeatmasker.org/cgi-bin/WEBRepeatMasker, accessed on 1 November 2024), revealing the presence of LTR/ERVL-MaLR elements (Figure 2A). Subsequently, we constructed overexpression vectors (Figure 2B). The full-length sequence of circRNA-0172 (1180 bp) was successfully cloned by us and amplified ~1000 bp flanking sequences upstream and downstream of the circRNA via PCR. These fragments were ligated into the pcDNA3.1 vector for transfection into DF-1 cells. Compared with the control group, we observed that transfection of the plasmid with flanking sequences significantly upregulated the expression of circRNA-0172 (Figure 2C), and at the same time, its host gene *MDM2* was also significantly upregulated (Figure 2D). This suggests that flanking sequences may enhance circRNA-0172 biosynthesis through mechanisms such as promoting back-splicing efficiency or increasing RNA stability and upregulating MDM2 expression. Collectively, sequences in intron 11, in the absence of intron 2, facilitate circRNA expression. these findings indicate that repeat sequences in the flanking introns regulate circRNA-0172 biogenesis.

### 3.3. Prediction of Potential Function of circRNA-0172

To further investigate the potential functions of circRNA-0172, this study employed Miranda (http://mirtoolsgallery.tech/mirtoolsgallery/node/1055, accessed on 5 May 2025) to predict its miRNA targets, identifying 85 candidate miRNAs. Among these, the top 6 miRNAs with the highest total energy scores were selected for further analysis. Using miRDB (https://mirdb.org/, accessed on 6 May 2025), TargetScan (https://www.targetscan.org/vert_80/, accessed on 6 May 2025), and DIANA (http://diana.imis.athena-innovation.gr/DianaTools, accessed on 6 May 2025) tools, 6268 potential target genes were initially predicted. After filtering genes with a cumulative weighted context++ score ≤ –0.5, 225 high-confidence target genes were finalized. Cytoscape 3.10.3 was used to construct a circRNA-0172-miRNA-mRNA regulatory network (Huaxi Biotechnology Co., Ltd., Beijing, China) (Figure 3), respectively, gga-miR-3524b-3p, gga-miR-1813, gga-miR-1596-5p, gga-miR-1680-5p, gga-miR-1586, gga-miR-1604, suggesting that circRNA-0172 may exert extensive gene regulatory effects through this network. To explore the potential relationship between circRNA-0172 and MDM2, we predicted MDM2-targeting miRNAs, yielding 87 candidates. A circRNA-0172-miRNA-MDM2 regulatory network (Figure 4) revealed that 10 of the 85 miRNAs predicted to target both circRNA-0172 and MDM2,respectively gga-miR-148a-3p, gga-miR-6623-3p, gga-miR-1150-3p, gga-miR-1760, gga-miR-6639-5p, gga-miR-1596-5p, gga-miR-1643-3p, gga-miR-1691, gga-miR-1655-5p, gga-miR-3524b-3p, suggesting that circRNA-0172 and MDM2 may jointly modulate the expression of downstream target genes via the ceRNA mechanism, thereby influencing immune pathways, physiological or pathological signals.

To explore the biological functions of circRNA-0172, GO and KEGG enrichment analyses were performed on predicted target genes. The GO analysis, integrating multi-dimensional data such as enrichment factors, *p*-values, and gene counts, visually demonstrated that circRNA-0172 target genes were highly enriched in intracellular, cytoplasmic, and organellar components (Figure 5). KEGG enrichment analysis revealed significant enrichment of target genes in pathways including glucose metabolism, fatty acid metabolism, protein processing, DNA replication, non-homologous end joining, cell cycle, apoptosis, autophagy, and differentiation, as well as cancer-related pathways such as PPAR signaling and p53 signaling pathways (Figure 6). These results provide critical functional annotations for deciphering the biological roles of circRNA-0172 at the levels of functional modules and pathway networks, laying an important foundation for subsequent mechanistic studies.

### 3.4. circRNA-0172 Promotes Apoptosis and Inhibits Proliferation

circRNA-0172 target genes were predicted to be associated with apoptosis and circRNA-0172 may involved in apoptosis. Next, we examined the effects of circRNA-0172 on cell viability and the expression of apoptosis-related genes. Cell viability assays confirmed that overexpression of circRNA-0172 reduced the viability of DF-1 cells and inhibited proliferation (Figure 7A). To further confirm the function of circRNA-0172, the expression levels of apoptosis-associated gene (*Bcl2*) and cell cycle-associated gene (*CCND1*) were detected via qRT-PCR. Overexpression of circRNA-0172 resulted in a significant downregulation of *Bcl2* gene expression (Figure 7B). Additionally, overexpression of circRNA-0172 led to a marked upregulation of *CCND1* gene expression (Figure 7C). Collectively, these results demonstrate that circRNA-0172 overexpression enhances apoptosis and suppresses proliferation of DF-1 cells.

## 4. Discussion

The molecular characteristics were systematically elucidated in this study, circularization mechanism, and potential functions of chicken circRNA-0172, it supplies new understandings of the molecular pathology related to ALV-J infection. ERVL-MALR repeat sequences constitute a specific family within the LTR-type repeat sequences, essentially representing genomic fragments derived from endogenous retrovirus-like elements.

CircRNA-0172 is a canonical exon-derived circRNA generated by back-splicing exons 3–11 of the *MDM2* gene. Its circular structure was validated via divergent primer PCR, sanger sequencing, and RNase R resistance assays, demonstrating exceptional stability distinct from linear RNAs. Circularization depends on repeat sequences in flanking introns. RepeatMasker analysis identified repetitive sequences within introns, ERVL-MALR repeat sequences constitute a specific family within the LTR-type repeat sequences, essentially representing genomic fragments derived from endogenous retrovirus-like elements, which may facilitate back-splicing by mimicking splice donor/acceptor sites or recruiting RNA-binding proteins of viral origin [41]. This exemplifies the host genome’s adaptive utilization of viral sequences.

Subcellular localization showed that circRNA-0172 was predominantly distributed in the cytoplasm, a key site for miRNA-target gene interactions [42], strongly suggesting its potential role via the ceRNA mechanism. In contrast, its host gene MDM2 mRNA was enriched in the nucleus, reflecting the strict “constraint” of post-transcriptional regulatory networks on proto-oncogene expression in normal cells [43]. Based on the ALV-J infection experiments and circRNA-0172 overexpression assays, this study hypothesizes that upon ALV-J infection, both circRNA-0172 and MDM2 are aberrantly inactivated. ALV-J may evade the host’s antiviral defenses by suppressing circRNA-0172 expression, potentially interfering with the immune signaling pathway or apoptotic mechanisms.

GO and KEGG enrichment analyses showed that predicted target genes were significantly enriched in pathways such as “DNA replication”, “cell cycle, apoptosis, autophagy and differentiation” and “p53 signaling pathway”. Among them, the p53 pathway is directly related to the proto-oncogenic function of *MDM2*. As a proto-oncogene, *MDM2* maintains cellular homeostasis by ubiquitinating and degrading p53, and its abnormal expression is closely associated with tumorigenesis [44].

Previous studies have shown that circRNA-0172 is differentially expressed in the spleens of black-boned chickens infected with ALV-J. Combined with the functional predictions of this study, it may act as a ceRNA to sponge miRNAs targeting *MDM2* (such as the predicted gga-miR-1596-5p and gga-miR-3524b-3p). Other prior studies have demonstrated that circRNAs can exert functions as ceRNAs, circFBXW4 acts as a ceRNA for miR-338-5p, competitively binding to miR-338-5p to relieve its inhibition of SLC5A7 expression, thereby upregulating SLC5A7 levels and ultimately inhibiting the progression of colorectal cancer [45]. During Kaposi’s sarcoma-associated herpesvirus infection, the virus-encoded vIRF1 protein activates the transcription of the host cell circARFGEF1 by binding to the transcription factor LEF1. circARFGEF1 acts as a ceRNA, binding to and degrading miR-125a-3p to relieve its inhibition of the target gene glutaredoxin 3, thereby promoting vIRF1-induced cell invasion and proliferations in vivo [46]. The circRNAome encoded by Epstein-Barr virus can act as a sponge for host miRNAs, sponging and inhibiting 56 potential miRNAs such as miR-15b-5p and miR-30c-1-3p to relieve their inhibition of target genes, thereby upregulating 1414 target genes and promoting cell cycle progression, tumorigenesis, and viral survival associated with EBV infection [47].

This study has several limitations: The interacting molecules of circRNA-0172 have only been predicted via bioinformatics; direct binding to miRNAs and ceRNA-mediated regulatory effects on *MDM2* require further validation; Functional studies are only based on overexpression plasmids, and knockdown models need to be constructed and combined with ALV-J infection to clarify its specific role in viral replication; In vivo mechanism studies are lacking, and chicken embryo or gene-edited animal models need to be used to explore the effect of circRNA-0172 on ALV-J oncogenicity. Future research can focus on: Identifying the direct interacting miRNAs and target genes of circRNA-0172; Evaluating its potential as an antiviral therapeutic target (such as designing circRNA ceRNA inhibitors).

## 5. Conclusions

This study provides the first evidence confirming that circRNA-0172 is a stable circRNA whose circularization is driven by repeat sequences in flanking introns. This finding not only expands the understanding of the diversity of circRNA circularization mechanisms but also furnishes potential targets for intervention in prevention and control of avian leukosis.

## Figures and Tables

**Figure 1 genes-16-01282-f001:**
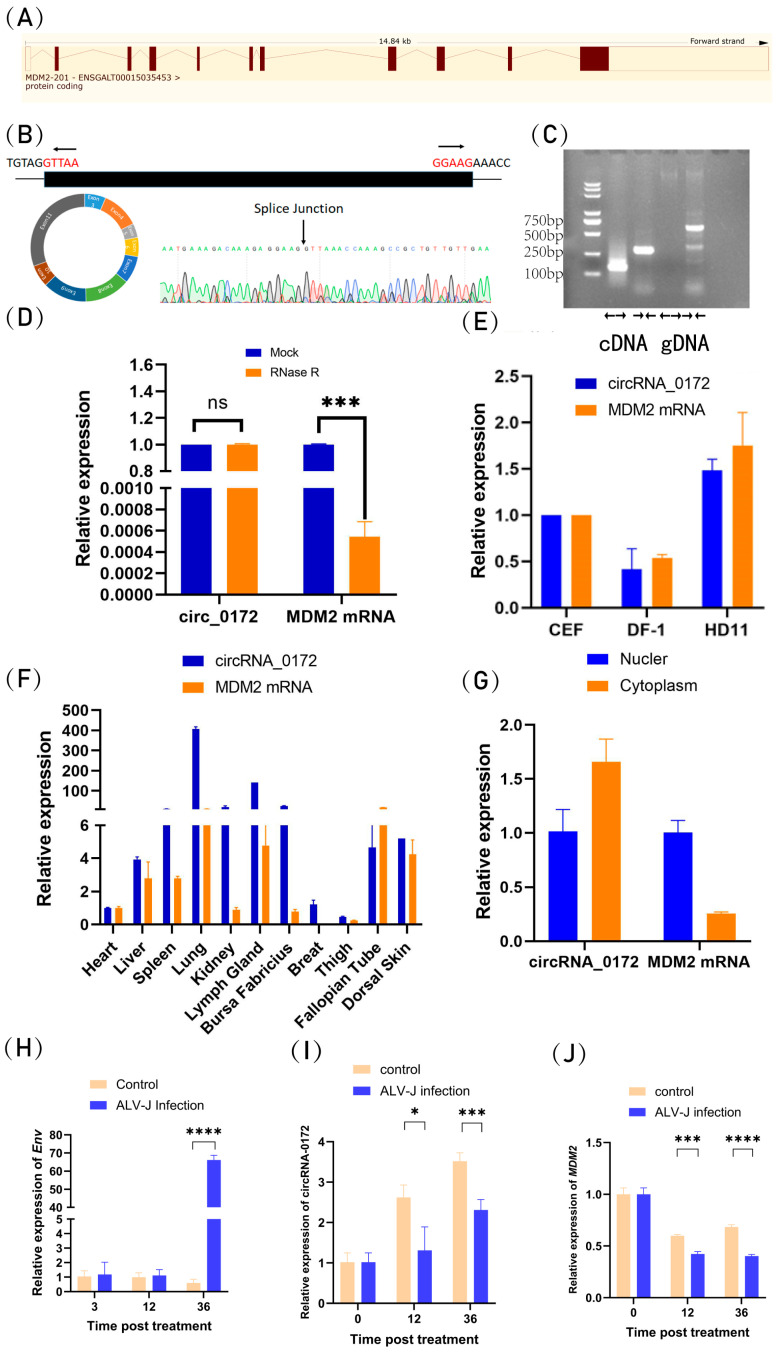
Characteristic analysis chicken-derived circRNA-0172. (**A**) circRNA-0172 is localized to exons 3–11 of the *MDM2* gene (full length 1180 bp). (**B**) Sanger sequencing. (**C**) Divergent primers and convergent primers confirming its circular structure. (**D**) RNase R treatment. (**E**) circRNA-0172 and *MDM2* mRNA cell expression profile. (**F**) circRNA-0172 and *MDM2* mRNA tissue expression profile. (**G**) circRNA-0172 and *MDM2* mRNA expression levels in the nucleus and cytoplasm. (**H**) *Env* expression levels after ALV-J virus infection. (**I**) circRNA-0172 expression levels after ALV-J virus infection. (**J**) *MDM2* expression levels after ALV-J virus infection. Data are presented as the mean ± SEM (*n* = 3). Statistical analysis was performed using *t*-test (for two groups) or one-way ANOVA (for multiple groups). (ns, not significant, * *p* < 0.05, *** *p* < 0.001, **** *p* < 0.0001).

**Figure 2 genes-16-01282-f002:**
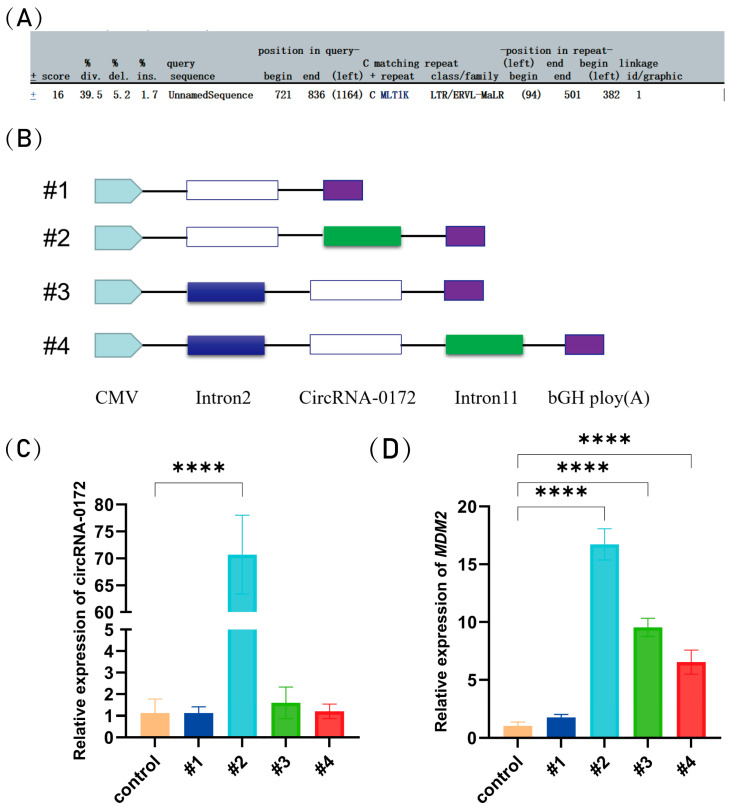
The Mechanism of Cyclisation of circRNA-0172. (**A**) RepeatMasker prediction. (**B**) Schematic of overexpression plasmids (#1–#4) containing different genomic sequences. (**C**) circRNA-0172 expression levels in DF-1 cells. (**D**) *MDM2* expression levels in DF-1 cells. Data are presented as the mean ± SEM (*n* = 3). Group differences were analyzed by one-way ANOVA (multiple-group comparisons). (**** *p* < 0.0001).

**Figure 3 genes-16-01282-f003:**
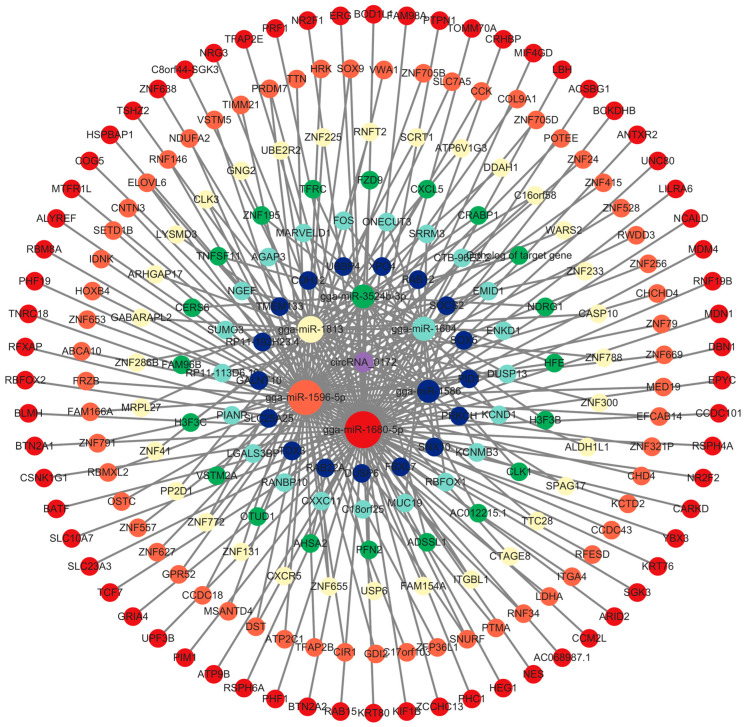
ceRNA (circRNA-0172-miRNA-mRNA) network diagram.

**Figure 4 genes-16-01282-f004:**
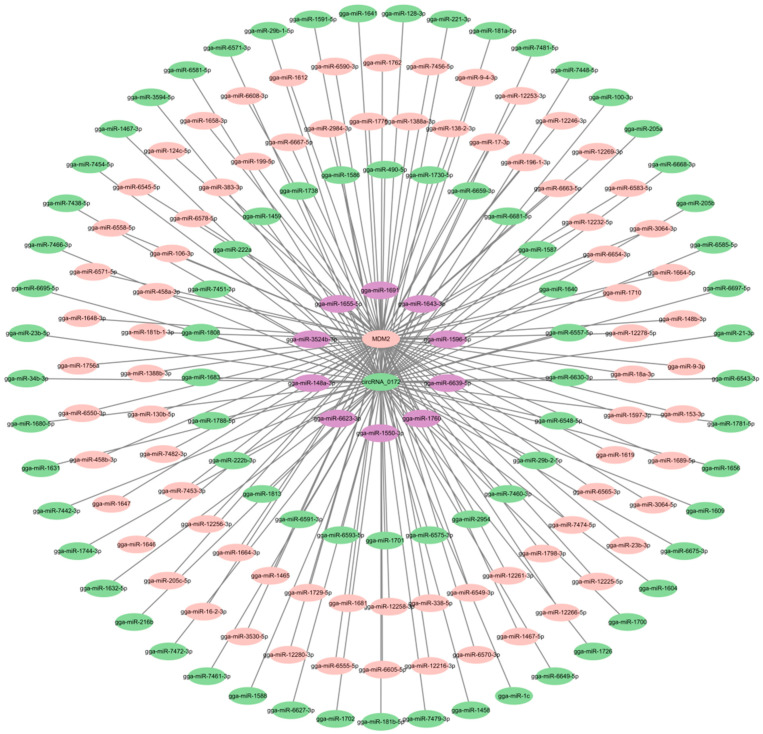
ceRNA (circRNA-0172-miRNA-*MDM2*) network diagram.

**Figure 5 genes-16-01282-f005:**
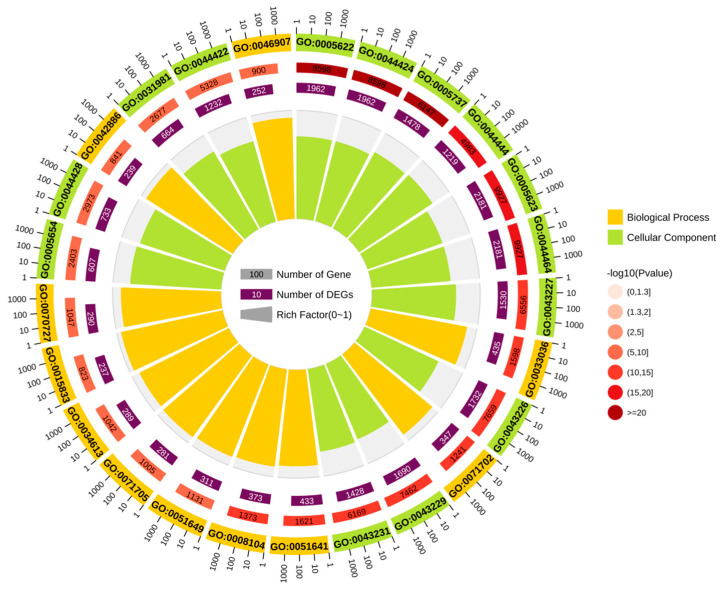
GO enrichment analysis of six miRNA predicted genes.

**Figure 6 genes-16-01282-f006:**
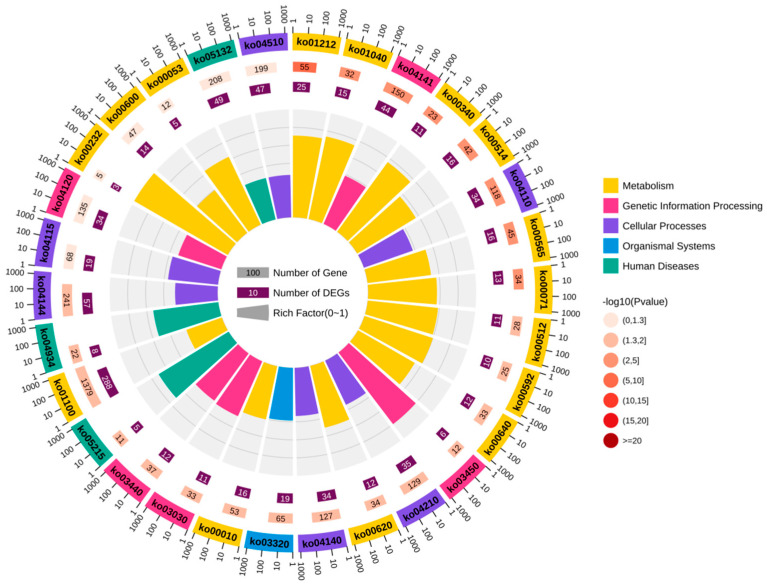
KEGG enrichment analysis of predicted target genes of 6 miRNAs.

**Figure 7 genes-16-01282-f007:**
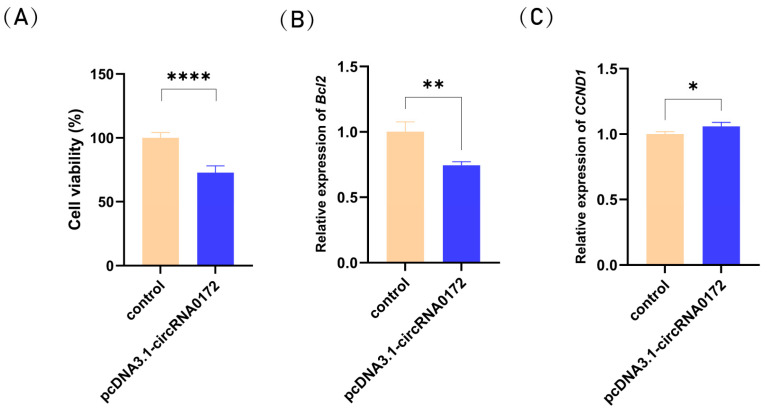
Effects of circRNA-0172 overexpression on cell apoptosis and proliferation. (**A**) Statistical analysis of cell viability. (**B**) *Bcl2* expression levels. (**C**) *CCND1* expression levels. Data are presented as the mean ± SEM (*n* = 3). Group differences were analyzed by one-way ANOVA (multiple-group comparisons). (* *p* < 0.05, ** *p* < 0.01 **** *p* < 0.0001).

**Table 1 genes-16-01282-t001:** Primer sequences employed in this study.

Primer Name	Primer Sequence (5′–3′)
circRNA-0172 (divergent primer)	F: CGAAGATTATTCTCAGCCAT R: CACCAGCTAACTTCAACAG
Convergent primer	F: GTTAAACCAAAGCCGCTGTTG R: CTCTTCCTCTTTGTCTTTCATTTCTTT
MDM2	F: CTCTACGCTGGCTGTTCC R: AATCGCTGCTATTGCTCC
GAPDH	F: CGATCTGAACTACATGGTTTAC R: TCTGCCCATTTGATGTTGC
#4-Intron2	F: agaGGAAGAGGTATGGAGTCCAGTCTGCCTGT R:aacgggccctctagactcgagAGTTCTTCATTTGTCTTTCCTGTTCACAT
#4-0172	F: CTCCACACAGGTTAAACCAAAGCCGCTGTTGTTG R: GACTCCATACCTCTTCCtctttgtctttcatttctttcttctcag
#4-Intron11	F: tagtccagtgtggtggaattcTGCGCTCCCCACGTGG R: TTGGTTTAACCTGTGTGGAGATGAAAACAGAAGAACAG
#3-Intron2	F: agaGGAAGAGGTATGGAGTCCAGTCTGCCTGT R:aacgggccctctagactcgagAGTTCTTCATTTGTCTTTCCTGTTCACAT
#3-0172	F: CTCCACACAGGTTAAACCAAAGCCGCTGTTGTTG R:AACGGGCCCTCTAGACTCGAGCTTCCtctttgtctttcatttctttcttctcag
#2-0172	F: tagtccagtgtggtggaattcGTTAAACCAAAGCCGCTGTTGTTGA R: GACTCCATACCTCTTCCtctttgtctttcatttctttcttctcag
#2-Intron11	F: tagtccagtgtggtggaattcTGCGCTCCCCACGTGG R: TTGGTTTAACCTGTGTGGAGATGAAAACAGAAGAACAG
#1-0172	F: tagtccagtgtggtggaattcGTTAAACCAAAGCCGCTGTTGTTGA R:AACGGGCCCTCTAGACTCGAGCTTCCtctttgtctttcatttctttcttctcag

## Data Availability

All datasets generated or analyzed during this study are available from the corresponding author on reasonable request.

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
