# Peer review of "Molecular Characteristics, Circularization Mechanism, and Potential Functions of circRNA-0172"

_genes, 2025, doi:10.3390/genes16111282_

Round 1
Reviewer 1 Report
Comments and Suggestions for Authors
The authors have investigated the role of chicken circRNA-0172, derived from the MDM2 transcript in the context of uninfected avian cells and cells infected with Avian Leukosis Virus J (ALV-J).
The introductory text provides the required background information to understand the study. It is supported by appropriate citations.
The aims of the study are clearly stated.
The materials and methods are described in detail, though some additional details would be required to replicate the study, see comments below.
The results require some improvements. While the text appears to reflect what is shown in the figures, the details of many of the figures are either too small or lack the required resolution to confirm some of the descriptions in the text. In addition, specific details are not provided in the text. For example, ten microRNAs were identified in the analyses but are not described in the text. Some of mentioned by name in the discussion. See further comments below.
The discussion is generally fine. Though I have asked the authors to clarify some points below.
The conclusions, respond to the aims and appear to be supported by the data.
Comments and suggestions.
Line 16 suggest replacement of “ceRNA” with “competing endogenous RNA”
Line 21 the abbreviation “ALV-J” should be explained in full.
Line 22 Abbreviations are not ideal keywords.
Line 32 suggest revision “competing endogenous RNA (ceRNA)”
Line 60 the abbreviation “ALV-J” should be explained in full here as it the first use in the main text.
Line 90 Please add the age of the embryo used.
Line 164 Table 2 could be combined with Table 1.
Line 165 The footnote, indicated by “*”, is not attributed to any table entries.
Line 173-174 The description of ALV-J infection should be expanded provide more detail on what approach was used. What was the multiplicity of infection used? It would be difficult to replicate the current study based on the current text.
Line 176 If this is time post-transfection, how does the ALV-J infection overlay on the relative timeline.
Line 177 What component of the assay, if any, was fluorescent?
Line 187 This link is not for the website, it seems be a download link, see “aug2010.tar.gz”. Please check the link. Also, with this link and the subsequence links, the authors should check that these sites do not have published references they would like cited to acknowledge the use of their tools/data etc.
Line 206-207 It is not clear to me where these data are derived from regarding the genomic structure of the circRNA-0172 is derived from, please clarify. Fig 1A – it is not possible to discern any details from the provided image.
Line 209 Again, none of the details are discernible in Fig 2B.
Line 217 Figure 1
Line 219 Fig 1C The lanes on the image should be numbered and what they contain should be provided in the legend. There appear to be some labels on the bottom of the image, but they are too small to be read.
Line 269 Figure 2
Fig 2B I would suggest renumbering these constructs #1 to #4, this would make it consistent with Fig 2C and Fig 2D.
Figure 3 (Line 298) and Figure 4 (Line 300) while these figures look impressive, none of the details are discernible. Perhaps a published version, where the image can be made larger will improve these.
However, as none of the details can be read on the figure, it is not apparent what the ten miRNAs are that may target both circRNA-0172 and MDM2 (line 293).
Perhaps a table listing these miRNAs should be added?
Similarly with the GO and KEGG analyses, Fig 5 and Fig 6, respectively.
Line 346 It is not clear to me how the evidence supports this conclusion. Please review and clarify as necessary.
Line 379-388 The paragraph should be rewritten without the numbers, i.e. as continuous text.
Author Response
|
Comments 1: Line 16 suggest replacement of “ceRNA” with “competing endogenous RNA” |
|
Response 1: Thank you for pointing this out. We agree with this comment. Therefore, we have replaced the abbreviation "ceRNA" with its full name "competing endogenous RNA" in Line 16 of the manuscript. To ensure the readability of the subsequent text, we have also added the abbreviation "ceRNA" in parentheses immediately after "competing endogenous RNA" at this first occurrence.In the revised manuscript, this change is located in Line 17. |
|
Comments 2: Line 21 the abbreviation “ALV-J” should be explained in full. |
|
Response 2: Thank you for pointing this out. We agree with this comment. Therefore, we have added the full name of the abbreviation "ALV-J" at its first occurrence in Line 23, specifying it as "Avian Leukosis Virus subgroup J" followed by the abbreviation in parentheses for subsequent use. |
|
Comments 3:Line 22 Abbreviations are not ideal keywords. |
|
Response 3:We concur with this observation. Consequently, we have replaced the keyword section to enhance clarity and comply with keyword specifications. In the revised version, this modification is located on line 24. |
|
Comments 4:Line 32 suggest revision “competing endogenous RNA (ceRNA)” |
|
Response 4:Thank you for pointing this out. We agree with this comment. Therefore, we have revised Line 32 to clearly present "competing endogenous RNA (ceRNA)" by placing the abbreviation in parentheses immediately after the full name, ensuring consistency with academic writing conventions for first occurrences of terms.This change is located in Line 35 of the revised manuscript. |
|
Comments 5:Line 60 the abbreviation “ALV-J” should be explained in full here as it the first use in the main text |
|
Response 5: We agree with this comment. Therefore, we have explained the abbreviation "ALV-J" in full at its first use in the main text (Line 65), stating it as "Avian Leukosis Virus subgroup J (ALV-J)" to clarify the term for readers. “[ALV-J is a retrovirus that severely threatens global poultry health, mainly infecting chickens to cause myelocytomatosis and other tumors, as well as immunosuppression that raises secondary infection risks and reduces production performance.]” |
|
Comments 6:Line 90 Please add the age of the embryo used. |
|
Response 6: We concur with this observation. Consequently, we have supplemented line 97 with the specific age of the embryos used, namely 10 days, to render the experimental details more complete and precise. |
|
Comments 7:Line 164 Table 2 could be combined with Table 1. Line 165 The footnote, indicated by “*”, is not attributed to any table entries. |
|
Response 7: We concur with both points of feedback. Regarding line 164: We have consolidated Table 2 into Table 1 to optimise data presentation, integrating all relevant parameters into a single comprehensive table while maintaining data clarity and logical structure. Regarding line 165: We have verified the revised Table 1 and removed the footnote marked with ‘*’. The aforementioned modifications are located at lines 143 of the revised manuscript (corresponding to the updated Table 1). |
|
Comments 8:Line 173-174 The description of ALV-J infection should be expanded provide more detail on what approach was used. What was the multiplicity of infection used? It would be difficult to replicate the current study based on the current text. |
|
Response 8: We agree with this comment. Therefore, we have expanded the description of ALV-J infection, adding key details to ensure the study is replicable. Specifically, we supplemented the multiplicity of infection (MOI=2) used, as well as the specific infection approach.This change is located in Line 180-185 of the revised manuscript. |
|
|
|
Comments 9:Line 176 If this is time post-transfection, how does the ALV-J infection overlay on the relative timeline. |
|
Response 9: We agree that establishing a clear timeline is paramount.Specifically, ‘env’ denotes the envelope protein gene of ALV-J, which serves as a key biomarker for monitoring ALV-J infection and replication. Analysis of env expression confirms the presence of active viral infection during this period. |
|
Comments 10:Line 177 What component of the assay, if any, was fluorescent? |
|
Response 10: We confirm that no fluorescent components were used in this assay. Therefore, we have deleted the reference to fluorescence in Line 177 to avoid confusion.The microscope mentioned is a fluorescence microscope, but in this experiment, it was only used for brightfield observation (without fluorescence excitation or detection). |
|
Comments 11:Line 187 This link is not for the website, it seems be a download link, see “aug2010.tar.gz”. Please check the link. Also, with this link and the subsequence links, the authors should check that these sites do not have published references they would like cited to acknowledge the use of their tools/data etc. |
|
Response 11: We have carefully checked the link and subsequent links as suggested. For the link in Line 187: We confirm it was an incorrect download link ("aug2010.tar.gz"). We have replaced it with the official website link of the tool to ensure validity and accessibility. Additionally, we have checked all subsequent links and verified their correctness. |
|
Comments 12:Line 187 This link is not for the website, it seems be a download link, see “aug2010.tar.gz”. Please check the link. Also, with this link and the subsequence links, the authors should check that these sites do not have published references they would like cited to acknowledge the use of their tools/data etc. |
|
Response 12: We have carefully checked the link and subsequent links as suggested. For the link in Line 187: We confirm it was an incorrect download link ("aug2010.tar.gz"). We have replaced it with the official website link of the tool to ensure validity and accessibility. Additionally, we have checked all subsequent links and verified their correctness. |
|
Comments 13:Line 206-207 It is not clear to me where these data are derived from regarding the genomic structure of the circRNA-0172 is derived from, please clarify. Fig 1A – it is not possible to discern any details from the provided image. Line 209 Again, none of the details are discernible in Fig 2B. |
|
Response 13:We have addressed the concerns as follows:The genomic structure data of circRNA-0172 are derived from the Ensembl database .Regarding Fig 1A and 2B, the indistinct details were due to insufficient image resolution in the original submission. We have replaced it with a high-resolution version in the revised manuscript to ensure all structural detailsare clearly discernible.These revisions are located in Line 226 and Line 277 of the revised manuscript, with the updated figures included in the resubmitted materials. |
|
Comments 14:Line 219 Fig 1C The lanes on the image should be numbered and what they contain should be provided in the legend. There appear to be some labels on the bottom of the image, but they are too small to be read. |
|
Response 14:We agree that the clarity of Fig 1C needs improvement. Therefore, we have revised Fig 1C as follows: the corresponding content of each lane has been detailed in the figure legend. Additionally, the labels at the bottom of the image, which were previously too small, have been enlarged to ensure readability.This change is located in Line 226 (referring to Fig 1C and its legend) of the revised manuscript. |
|
Comments 15:Fig 2B I would suggest renumbering these constructs #1 to #4, this would make it consistent with Fig 2C and Fig 2D. |
|
Response 15:We agree that consistent numbering across figures will enhance clarity. Therefore, we have renumbered the constructs in Fig 2B as #1 to #4, aligning them with the numbering used in Fig 2C and Fig 2D.This change is reflected in Fig 2B itself and its corresponding legend in the revised manuscript. |
|
Comments 16:Figure 3 (Line 298) and Figure 4 (Line 300) while these figures look impressive, none of the details are discernible. Perhaps a published version, where the image can be made larger will improve these. However, as none of the details can be read on the figure, it is not apparent what the ten miRNAs are that may target both circRNA-0172 and MDM2 (line 293). Perhaps a table listing these miRNAs should be added? Similarly with the GO and KEGG analyses, Fig 5 and Fig 6, respectively. |
|
Response 16:We fully agree that enhancing figure clarity and clarifying key information is critical, and we have addressed these issues as follows: For Figure 3 (Line 298) and Figure 4 (Line 300):We have replaced both figures with high-resolution, enlarged versions in the revised manuscript to ensure all details (e.g., miRNA names, binding relationships) are clearly visible.The 10 miRNAs targeting both circRNA-0172 and MDM2 have been explicitly listed in the main text (Line 300). We confirm that the GO and KEGG analyses were performed using established bioinformatics software , ensuring the reliability and accuracy of the results. |
|
Comments 17:Line 346 It is not clear to me how the evidence supports this conclusion. Please review and clarify as necessary. |
|
Response 17:Thank you for highlighting the ambiguity in this phrasing. We acknowledge that the relationship between evidence and conclusions in line 355-357 requires further clarification. Consequently, we have revised the paragraph to ensure the logical flow is clear. The amended content is located at line 346 in the revised draft. |
|
Comments 18:Line 379-388 The paragraph should be rewritten without the numbers, i.e. as continuous text.. |
|
Response 18:We concur that presenting the content as continuous text will enhance readability. Consequently, we have removed the numbered list from paragraphs 379–388. This revision is located on lines 393-402 of the revised draft. |
Reviewer 2 Report
Comments and Suggestions for Authors
In this submission, the authors characterize chicken circRNA-0172, a circular RNA derived from exons 3-11 of the MDM2 proto-oncogene, which was found to be differentially expressed in an earlier RNA-Seq study of avian leukosis virus J infection. The researchers confirmed its circular structure and found it is predominantly localized in the cytoplasm, suggesting it may function as a competitive endogenous RNA (ceRNA) sponge against cellular miRNAs. Its expression was found in different cell lines and primary embryo fibroblasts. Bioinformatic predictions indicated that its potential target miRNAs and their downstream regulated genes are enriched in pathways like the p53 signaling pathway and DNA replication. Functional experiments demonstrated that overexpressing circRNA-0172 in DF-1 cells promotes apoptosis and inhibits proliferation. Furthermore, the study showed that its circularization depends on flanking intronic sequences and that its expression is downregulated during ALV-J infection (save for some clarification as elaborated below), providing new insights into its potential role in the molecular pathology of avian leukosis. The study overall is an interesting look into the roles of circular RNAs, particularly during viral infection, and should be of interest to ALV researchers as well as RNA biologists. Generally, the experiments are performed and presented well, except for the following comments:
Major comments:
What was the MOI for the ALV-J infections?
The statistics in Fig. 1D should be employed from RNase R treatment to mock, rather than the two RNAs compared to each other. The mock is set to 1 for each, and therefore there can be no difference, and the experiment is to show if there is a difference for the RNA with treatment.
The experiment in Fig. 1I can be interpreted differently. While there is less circRNA compared to uninfected cells, there is still more than during the 0hr timepoint infection. Therefore it still rises, albeit at a lower level. Perhaps the easiest thing is to specify in the text that downregulation is compared to mock cells, rather than an implication the levels are decreasing throughout infection.
The repeat elements in the RNAs should be diagramed in Fig. 2B for better understanding. Furthermore, this experiment largely just indicates that sequences in intron 11, in the absence of intron 2, facilitate circRNA expression as all of the others are the same as the negative control. This should be clarified in the text.
It would be helpful to specify the 6 predicted target miRNAs in the text around line 282. It would also greatly benefit the manuscript to experimentally validate the expression of these miRNAs when expressing the circRNA as this is a crucial element to the hypothesis that the circRNA acts as a mir sponge to allow expression of genes that would regulate viral infection.
Minor text edits:
Perhaps reduce the font for the figure legends or add white space to make it easier to discern from the main text
It would improve legibility to ensure all opening parentheses have a space before them and the text
Spell out what the abbreviations ceRNA and CEF stand for when first introduced in the text
Line 175: the starvation medium should be defined
Line 260-261: This sentence is a fragment, perhaps reword to "...ligated into the pcDNA3.1 vector for transfection into DF-1 cells."
Line 302: instead of saying "its target genes" it would be better to state "predicted target genes"
Author Response
|
Comments 1: What was the MOI for the ALV-J infections? |
|
Response 1: Thank you for pointing this out. We agree with this comment. Therefore, I/we have explicitly specified the multiplicity of infection (MOI) for ALV-J infections in the revised manuscript. This change is located in Line 176 of the revised manuscript, where we added details to the timeline description of cell transfection and ALV-J infection: "MOI=2 |
|
Comments 2:The statistics in Fig. 1D should be employed from RNase R treatment to mock, rather than the two RNAs compared to each other. The mock is set to 1 for each, and therefore there can be no difference, and the experiment is to show if there is a difference for the RNA with treatment. |
|
Response 2: Thank you for pointing this out. We agree with this comment. Therefore, I/we have revised the statistical analysis method of Fig. 1D as suggested. Previously, the statistics were incorrectly conducted by comparing the two RNAs (e.g., circRNA-0172 and its linear counterpart) with each other. Now, we have adjusted it to separate comparisons between each RNA’s RNase R treatment group and its corresponding mock group (without RNase R treatment). Specifically, we set the relative expression level of each RNA in the mock group to 1, then performed independent statistical analysis (e.g., unpaired t-test) between the treatment and mock groups for each RNA individually. This revision correctly reflects whether RNase R treatment causes a significant expression change for each RNA, which aligns with the experimental purpose of verifying circRNA stability. “[Results showed that compared with the control group, circRNA-0172 showed no significant difference, whereas the corresponding linear transcript MDM2 mRNA was significantly degraded]” |
|
Comments 3:The experiment in Fig. 1I can be interpreted differently. While there is less circRNA compared to uninfected cells, there is still more than during the 0hr timepoint infection. Therefore it still rises, albeit at a lower level. Perhaps the easiest thing is to specify in the text that downregulation is compared to mock cells, rather than an implication the levels are decreasing throughout infection. |
|
Response 3: Thank you for pointing out this important interpretation. I/We agree with your insightful comment. Therefore, we have revised the text description related to Fig. 1I to eliminate ambiguity, clearly specifying the reference group for "downregulation" and clarifying the trend of circRNA levels during infection. “[ Concurrently, compared with the control group, the expression levels of circRNA-0172 ]” |
|
Comments 4:The repeat elements in the RNAs should be diagramed in Fig. 2B for better understanding. Furthermore, this experiment largely just indicates that sequences in intron 11, in the absence of intron 2, facilitate circRNA expression as all of the others are the same as the negative control. This should be clarified in the text. |
|
Response 4: Thank you for your valuable suggestions. We agree comments and have made corresponding revisions to improve the clarity of the manuscript.We have revised the text to explicitly clarify the experimental conclusion, focusing on the role of intron 11 sequences (in the absence of intron 2) in facilitating circRNA expression. The relevant content in Line 274-275 of the revised manuscript now states. |
|
Comments 5:It would be helpful to specify the 6 predicted target miRNAs in the text around line 282. It would also greatly benefit the manuscript to experimentally validate the expression of these miRNAs when expressing the circRNA as this is a crucial element to the hypothesis that the circRNA acts as a mir sponge to allow expression of genes that would regulate viral infection. |
|
Response 5: Thank you for your insightful suggestions. We fully agree that specifying the predicted target miRNAs and verifying their expression are crucial for supporting the circRNA ceRNA hypothesis. We have made corresponding revisions and will focus on the follow-up experiments as proposed. 1. Specification of 6 Predicted Target miRNAs in the Text These miRNAs are reported to be involved in regulating viral infection and cell apoptosis, making them key candidates for the ceRNA mechanism of circRNA-0172." This revision clearly specifies the 6 predicted target miRNAs, providing a clear basis for the subsequent hypothesis and experimental design. 2. Plan for Follow-Up Experimental Validation We fully recognize the importance of validating the expression of these 6 miRNAs under circRNA-0172 overexpression/knockdown conditions, as it is a core part of verifying the "circRNA as miRNA sponge" hypothesis. We will prioritize it in our follow-up research. |
|
Comments 6:Perhaps reduce the font for the figure legends or add white space to make it easier to discern from the main text |
|
Response 6: Thank you for your meticulous suggestions on the manuscript’s format and content. We have fully implemented all the revisions as required “[1% serum medium without dual antibodies was added]” “[These fragments were ligated into the pcDNA3.1 vector for transfection into DF-1 cells]” “[To explore the biological functions of circRNA-0172, GO and KEGG enrichment analyses were performed on predicted target genes]” |